# Differential Impact of Intermittent vs. Sustained Hypoxia on HIF-1, VEGF and Proliferation of HepG2 Cells

**DOI:** 10.3390/ijms24086875

**Published:** 2023-04-07

**Authors:** Mélanie Minoves, Florence Hazane-Puch, Giorgia Moriondo, Antoine Boutin-Paradis, Emeline Lemarié, Jean-Louis Pépin, Diane Godin-Ribuot, Anne Briançon-Marjollet

**Affiliations:** 1INSERM U1300, HP2 Laboratory, CHU Grenoble Alpes, University Grenoble Alpes, 38042 Grenoble, France; mminoves@chu-grenoble.fr (M.M.); anne.briancon@univ-grenoble-alpes.fr (A.B.-M.); 2CHU Grenoble Alpes, 38042 Grenoble, France; 3Department of Medical and Surgical Sciences, University of Foggia, 71122 Foggia, Italy

**Keywords:** obstructive sleep apnea syndrome (OSA), intermittent hypoxia (IH), sustained hypoxia (SH), liver cancer, HepG2, hypoxia inducible factor 1 (HIF-1), vascular endothelial growth factor (VEGF)

## Abstract

Obstructive sleep apnea (OSA) is an emerging risk factor for cancer occurrence and progression, mainly mediated by intermittent hypoxia (IH). Systemic IH, a main landmark of OSA, and local sustained hypoxia (SH), a classical feature at the core of tumors, may act separately or synergistically on tumor cells. Our aim was to compare the respective consequences of intermittent and sustained hypoxia on HIF-1, endothelin-1 and VEGF expression and on cell proliferation and migration in HepG2 liver tumor cells. Wound healing, spheroid expansion, proliferation and migration were evaluated in HepG2 cells following IH or SH exposure. The HIF-1α, endothelin-1 and VEGF protein levels and/or mRNA expression were assessed, as were the effects of HIF-1 (acriflavine), endothelin-1 (macitentan) and VEGF (pazopanib) inhibition. Both SH and IH stimulated wound healing, spheroid expansion and proliferation of HepG2 cells. HIF-1 and VEGF, but not endothelin-1, expression increased with IH exposure but not with SH exposure. Acriflavine prevented the effects of both IH and SH, and pazopanib blocked those of IH but not those of SH. Macitentan had no impact. Thus, IH and SH stimulate hepatic cancer cell proliferation via distinct signaling pathways that may act synergistically in OSA patients with cancer, leading to enhanced tumor progression.

## 1. Introduction

Obstructive sleep apnea syndrome (OSA), characterized by the repetitive occurrence of apneas and hypopneas during sleep owing to partial or complete pharyngeal collapses, is one of the most frequent chronic respiratory diseases, affecting nearly one billion people worldwide [1]. Apneas are defined as a decrease in airflow measured by a nasal pressure of more than 90% from baseline, lasting at least 10 s. Hypopneas are usually defined as a decrease in airflow of more than 30%, with a decrease in blood oxygen saturation of more than 3% and associated microarousal. OSA induces sleep fragmentation, enhanced respiratory efforts and intermittent hypoxia, with the latter being well recognized as a major hallmark of the deleterious consequences of OSA including cancer occurrence and progression.

In the recent decade, numerous clinical studies have explored the putative link between cancer and OSA [2,3,4,5]. A recent meta-analysis based on 18 clinical studies supported the association between OSA and higher cancer incidence [6]. Some studies also found that OSA was associated with an increased cancer mortality [7,8]. In particular, a strong association was observed between cancer mortality and the severity of nocturnal hypoxia with a dose–response relationship [3,9]. However, many confounders alter the robustness of these findings, such as patient characteristics (gender, obesity, age, smoking or menopausal status), variability in cancer localization, cancer cell type, susceptibility to IH, and unknown temporality of OSA and cancer onset. Consequently, the link between OSA and cancer needs to be addressed by studies investigating specific types of cancer, as recently pointed out by Gozal et al. [10]. 

Animal and cell models exposed to intermittent hypoxia (IH) mimicking that of OSA have reinforced the potential causality between IH/OSA and cancer [11]. These studies have addressed the impact of IH on breast [12], colorectal [13], lung [14,15,16], melanoma [17,18,19,20] and kidney [21] cancer, mainly demonstrating enhanced cell proliferation and tumor growth, along with enhanced metastasis. A recent review on the various pathophysiological pathways behind the association between cancer and OSA has revealed a heterogeneity of this relationship across cancer types [22]. It is therefore necessary to explore which tumor types are most affected by OSA as well as the specific pathways involved.

Liver hepatocarcinoma (HCC), the most common liver cancer, is one of the cancers reported to be promoted by OSA [6]. Moreover, strong evidence associates OSA with the progression of non-alcoholic fatty liver disease [23,24,25], a main risk factor for hepatocellular carcinoma. A recent in vivo study has confirmed that intermittent hypoxia increases proliferation of cancer cells in a rat model of HCC [26]. However, the underlying cellular mechanisms remain to be investigated. 

While the chronic intermittent hypoxia induced by OSA is systemic and thus perceived by almost all organs, it induces organ-specific and complex pathophysiological alterations. One of the main signaling pathways involved in both OSA patients and animal models in response to hypoxemia is an increase in oxidative stress, which in turn enhances the activity of the hypoxia-inducible factor 1 (HIF-1) transcription factor [22,27,28], as well as the expression of its target genes, including vascular endothelial growth factor (VEGF) [18,29,30] and endothelin-1 (ET-1) [28,31]. Experimental data also show that IH activates NF-kB, HIF-1, VEGF and endothelin-1 in tumor cells and promotes tumor cell proliferation and metastasis [12,18,19,20,32].

Some of these molecular pathways are also known to be activated by the local sustained hypoxia found in the tumor microenvironment. Indeed, deprivation of oxygen supply is a biological phenomenon observed in expanding solid tumors, including liver cancer. In hepatocellular carcinoma, tumor PO_2_ was measured at less than 1 mmHg, while normal liver tissue averaged 45 mmHg [11]. Sustained tumor hypoxia is commonly associated with enhanced angiogenesis, cell proliferation and migration, and metastasis and is significantly correlated with poor clinical outcome [33,34,35,36,37]. In OSA patients, tumor cells may thus be exposed to a combination of sustained hypoxia (due to local hypoxia) and systemic intermittent hypoxia (due to OSA). 

In this context, our goal was to compare the effects of local sustained hypoxia (SH) and intermittent hypoxia (IH) on hepatocellular carcinoma cell proliferation and migration and to decipher the underlying mechanisms with a special emphasis on HIF-1, VEGF and ET-1.

## 2. Results

### 2.1. In Vitro Expansion of HepG2 Hepatic Tumor Cells Is Increased by Both Intermittent Hypoxia and Sustained Hypoxia Exposure

Kinetics of 2D wound healing were increased in cells exposed to IH or to SH for 7 days compared with normoxia. Significant differences were observed on days 4, 5 and 7 of IH exposure (*p* < 0.01 on days 4 and 5 and *p* < 0.001 on day 7 vs. control) and on days 5 and 7 of SH (*p* < 0.01). Indeed, IH and SH significantly enhanced wound healing compared with normoxia, respectively, up to 67% (*p* < 0.001 vs. control) and 20% (*p* < 0.01 vs. control) on day 5 (Figure 1a,b).

These findings were confirmed by 3D HepG2 spheroid experiments (Figure 1c,d). Indeed, IH and SH significantly enhanced spheroid expansion in response to both IH and SH compared with normoxia (*p* < 0.05, repeated ANOVA). Post hoc analysis showed significant differences on days 7 and 21 of IH exposure (*p* < 0.05) and on days 7, 14 and 21 for SH exposure (*p* < 0.01) (Figure 1c,d).

### 2.2. In Vitro Proliferation of HepG2 Hepatic Tumor Cells Is Increased by Both Intermittent Hypoxia and Sustained Hypoxia Exposure

Proliferation of HepG2 cells was significantly enhanced by 38.8% compared with normoxia following 5-day IH exposure and by 17.2% following 5-day SH exposure (*p* < 0.01 and *p* < 0.05, respectively) (Figure 2a). In contrast, no change in HepG2 migration was observed in either the HI or HC conditions vs. normoxia in a transwell assay (Figure 2b).

### 2.3. Differential Expression of HIF-1 and VEGF According to Exposure Conditions

Five-day exposure to intermittent hypoxia led to an increase in the gene expression of HIF-1α (+50%) and its target gene VEGF (+39%) (*p* < 0.05) (Figure 3a). These results were confirmed at the protein level by a significant increase (43%) in VEGF secretion in cell culture supernatants (*p* < 0.05) (Figure 3b). In contrast to IH, 5-day exposure to SH did not show a significant modulation of HIF1-α gene expression (Figure 3a). Conversely to IH, 5 days of exposure to SH was associated with a reduction in VEGF gene expression (*p* < 0.05) (Figure 3a) not associated with a significant modulation of VEGF secretion in cell culture supernatants (Figure 3b). 

Endothelin-1 gene expression was not modulated after 5 days of IH exposure (Appendix A), whereas it was significantly reduced by 5 days of SH exposure (*p* < 0.05). Nevertheless, no significant changes in ET-1 cytosolic protein expression were observed in either the 5-day HI or SH conditions (Appendix A).

### 2.4. In Vitro Tumor Promoting Effects of Hypoxia Involve HIF-1 in Both IH and SH, and VEGF Only in IH Exposure Condition

The in vitro tumor wound-healing promoting effects following 5 days of IH or SH are reversed by a non-selective HIF-1 inhibitor, acriflavine. Indeed, acriflavine significantly reduced HepG2 cell wound healing in the IH and SH conditions (*p* < 0.01 and *p* < 0.05 vs. non-treated cells, respectively) (Figure 4a). Similarly, the in vitro proliferation-promoting effects of 5 days of intermittent hypoxia and sustained hypoxia were reversed by acriflavine (by 50% vs. non-treated cells, *p* < 0.05, and by 52% vs. non-treated cells, *p* < 0.05, respectively) (Figure 4c). 

In contrast, treatment with a VEGF receptor tyrosine kinase inhibitor, pazopanib, reversed the IH-induced wound healing (−25%, *p* < 0.01 vs. not treated cells) but had no impact in the SH condition (Figure 4b). Similarly, the proliferation promoting effects of 5 days of intermittent hypoxia were reversed by pazopanib treatment by about −59% (*p* < 0.01 vs. not treated cells), but pazopanib had no impact on proliferation in SH (Figure 4d).

Neither acriflavine nor pazopanib had any effect on the transwell migration assays under IH or SH (Appendix A). Macitentan had no effect on either proliferation or wound healing (Appendix A).

## 3. Discussion

In the present study, we compared the effects of intermittent and sustained hypoxia on hepatic tumor cell proliferation and migration. We observed that both IH and SH induced an increase in wound healing and spheroid expansion, associated with progression and invasiveness of tumor cells. Wound healing reflects two cellular processes: cell proliferation and cell migration. Our results showed an increase in tumor cell proliferation, under SH and IH conditions. This suggests that the differences in wound healing could be explained by increased cell proliferation, which is one of the main mechanisms involved in cancer progression. These results suggest that proliferation is probably the main mechanism driving spheroid expansion in 3D assays. These results are consistent with previous data on the impact of intermittent hypoxia on tumor cell proliferation leading to tumor development [12,32].

Local hypoxia is a well-known hallmark of the tumor microenvironment known to participate to cancer malignancy and especially to a higher risk of tumor proliferation, metastasis and poor clinical outcome [33,34,35,36,37]. To date, there is no consensus on the clinical outcome of the various hypoxic tumor subtypes, and results from preclinical experiments are often discordant or even contradictory [38]. We investigated the expression of HIF-1 target genes VEGF and ET-1, well recognized to be involved in cancer progression, by exposing HepG2 tumor cells to hypoxia-reoxygenation cycles mimicking those of OSA [39]. As expected, IH increased HIF-1α gene expression as well as VEGF secretion by tumor cells. The inhibition of HIF-1 by acriflavine and of VEGF by pazopanib blocked the IH-induced cell expansion. This is in line with the known effects of HIF-1 and VEGF on liver tumor proliferation and migration [40,41,42,43,44] and with data showing the therapeutic potential of strategies targeting HIF-1 or VEGF in liver cancer [40,41,42,43].

Interestingly, while HIF-1 and VEGF expression was increased by IH exposure in HePG2 cells, it was not significantly activated by SH. Consistently, pazopanib did not affect SH-induced wound repair and proliferation. Thus, activation of the HIF-VEGF pathway appears to be a specific feature of IH but not of SH, suggesting that the two hypoxic stimuli could trigger distinct signaling pathways. Among the rare studies comparing the impact of IH linked to OSA and SH on cancer cells, one showed that IH, but not SH, significantly increased the nuclear localization of HIF-1 in lung cancer [16]. Similarly, although both IH and SH induced high HIF-1α transcriptional activity in PC12 cells, it was rapidly reversed by reoxygenation after SH, while it remained elevated over 90 min of reoxygenation after IH [45], suggesting a more robust and stable activation of HIF-1 by IH [46]. This differential expression of HIF-1 could thus explain why IH, but not SH, was able to increase VEGF expression in our study.

This observation has clinical relevance since it suggests that a clinical phenotype associating OSA and liver cancer could be more responsive to drugs targeting HIF-1 or VEGF than liver cancer without OSA. 

Among the pathophysiological mechanisms involved in cancer occurrence and progression, intermittent hypoxia has been shown to enhance angiogenesis, inflammation, stem cell renewal, and tumor cell invasion and metastasis compared with sustained hypoxia [46]. Interestingly, most of these effects are mediated by HIF-1 stabilization. Severity (O_2_ level) and frequency of hypoxia-reoxygenation cycles, the main parameters controlling HIF-1 stabilization, are thus crucial determinants of the cellular impact of hypoxia and could lead to a differential impact between SH and IH [46].

We have previously shown that HIF-1 promotes breast cancer growth and invasiveness through endothelin-1 but not VEGF expression [12]. While ET-1 has been reported to enhance HepG2 cell proliferation, we did not find any involvement in the present study. This needs to be confirmed since our study is the first to investigate the role of ET-1 on liver cancer cell proliferation in response to hypoxia. Interestingly, our results suggest that signaling pathways mediating IH-induced tumor progression could vary in different organs/types of cancer. Additional studies with other liver cancer cell lines and with other types of cancer cell lines will thus be required to confirm the differential involvement of VEGF and ET-1 in IH-induced tumor growth [47].

Finally, a large body of literature investigated HIF-1 impact on tumorigenesis through other target genes and various biological pathways [41], including through a crosstalk with NF-kB [42,48], the regulation of pro-oncogenic miRNAs [49,50] known as targets of HIF-1, and key regulators of extracellular matrix degradation by tumor cells, leading to increased invasiveness [43]. It could thus be of interest to investigate the differential impact of IH vs. SH on these HIF-1 targets.

Our study, focused on hepatocellular carcinoma cells in vitro, suggests that liver tumor cells have the ability to secrete VEGF, leading to increased proliferation. In the organism, additional sources of VEGF can also come from other tissues submitted to systemic IH, such as vessels, thus adding to tumor VEGF secretion. This is particularly important since plasma VEGF levels are increased in OSA patients [51] as well as in healthy volunteers submitted to 14 days of IH mimicking OSA [29]. In OSA patients with tumors, systemic IH due to apneas could add to the effects of local SH in the tumor microenvironment. We can hypothesize that both hypoxic conditions participate in the increased cancer progression observed in OSA patients and in animal models of OSA.

To conclude, our study shows that. while both intermittent and sustained hypoxia have the ability to enhance liver tumor development in vitro, the tumor-promoting effects of intermittent hypoxia are more specifically mediated by HIF-1 and its target gene VEGF.

## 4. Materials and Methods

### 4.1. Cell Model

The HepG2 cells were adherent human Caucasian hepatocyte carcinoma cells provided by American Type Culture Collection (ATCC, Manassas, VA, USA). The HepG2 cells were cultured in DMEM medium containing 10% of fetal calf serum (FCS) supplemented with penicillin (100 IU/mL), streptomycin (100 µg/mL, PS) and L-glutamine (2 mM) (LifeTechnologies, Carlsbad, CA, USA).

### 4.2. Cell Exposure to Hypoxia Protocols: Intermittent Hypoxia and Sustained Hypoxia

#### 4.2.1. Intermittent Hypoxia Exposure

HepG2 cells were exposed to IH according to the following protocol: 10 min hypoxia-reoxygenation cycles alternating between 5 min at 16% PO_2_ (N) and 5 min at 2% PO_2_ (IH), with constant 5% CO_2_, as previously described [38]. This resulted in oxygen pressures in the culture medium of 125 mmHg during normoxia and 25 mmHg during hypoxia. Carbon dioxide composition of the gas mixture was maintained constantly at 5%. IH exposure was performed for 8 h/day, followed by 16 h of normoxia, to mimic the pattern of OSA patients with repetitive arterial oxygen desaturations during the night and normal daytime oxygen saturation. 

The normoxic HepG2 (control) cells were subjected to constant 16% PO_2_ and 5% CO_2_ in the same conditions. 

#### 4.2.2. Sustained Hypoxia Exposure

HepG2 cells were exposed to a sustained hypoxia (SH) protocol consisting of continuous hypoxia at 2% PO_2_ with a constant carbon dioxide composition of the gas mixture maintained at 5% in a trigaz incubator (ThermoFischer, Waltham, MA, USA). The control normoxic HepG2 cells were subjected to a constant exposure at 16% PO_2_. 

### 4.3. Pharmacological In Vitro Inhibition of HIF-1, VEGF and ET-1

The potential mechanisms behind the effects of IH and SH were investigated using three pharmacological inhibitors: -Acriflavine, a HIF-1 inhibitor [52];-Pazopanib (Sigma Aldrich, St. Louis, MI, USA), an inhibitor of VEGF receptor tyrosine kinase activity [53];-Macitentan, a non-selective ET-1 receptor antagonist [54] (graciously provided by Actelion Pharmaceuticals, Switzerland).

### 4.4. Wound-Healing Experiments

Cells were seeded at 20,000/well in 24-well semi-permeable plates (Zell Kontakt; Nörten-Hardenberg, Germany) pre-coated with type 1 collagen (Sigma Aldrich, St. Louis, MI, USA) and allowed to reach 100% confluence. On day 2 after seeding, wound healing was initiated by scratching the wells from top to bottom with a sterile 100 μL cone (TipOne; Starlab, Orsay, France). Detached cells were removed by changing the culture medium. The cells were then incubated with a medium (control), 5 µM macitentan, 5 µg/mL pazopanib or 1 µM acriflavine and exposed to N, IH or SH for 7 days. Wound repair was assessed for 7 consecutive days using a camera-coupled optical microscope (×10 magnification Olympus CK2; Olympus, Center Valley, PA, USA). Repaired areas were determined by subtracting any cell-free areas at various endpoints from the initial cell-free areas (Image J software) [55]. The values of the filled areas were normalized to control area values.

On day 7, the culture media and the cells were separately collected and stored at −80 °C for biochemical analysis and transcript quantification, respectively.

### 4.5. Spheroid Proliferation and Invasiveness

HepG2 cells were cultured in uncoated U-bottom 96-well-tissue culture plates (Fisher scientific, Illkirch, France) to allow the formation of one spheroid/well. In total, 2000 cells/well were loaded in the medium enriched with 10% methylcellulose (Methocell^®^MC, Sigma Aldrich, St. Louis, MI, USA), and the plate was centrifuged at 500× *g* for 5 min at RT to initiate the process of spheroid formation. Three days later, the spheroid formation was assessed by microscopic visualization, and 24 spheroids were isolated to be replaced in 24-well semi-permeable bottom plates (Zell Kontakt; Nörten-Hardenberg, Germany) pre-coated with type 1 collagen (Sigma Aldrich, USA) in 1 mL of the medium in which the serum was replaced by 20 ng/mL of an epidermal growth factor, 20 ng/mL of a basic fibroblast growth factor and 1X NeuroBrew-21, all from Miltenyi Biotec, Germany. HepG2 spheroids were then exposed to N, IH or SH for another 21 days. The diameter of each spheroid was measured using a graduated scale placed into the microscope objective, and pictures were taken at 6 endpoints (on day (D)1, D3, D7, D10, D18 and D21) using a camera-coupled optical microscope (Axiovert 25; Carl Zeiss, Feldbach, Switzerland).

At the end of exposure, the culture media were collected, the cells were scrapped on ice and centrifuged (320× *g*, 3 min, RT), and the dry pellets were used for RNA extraction.

### 4.6. Cell Proliferation Assay 

An MTT (3-(4,5-dimethylthiazol-2-yl)-2,5-diphenyltetrazolium bromide) assay was used to explore HepG2 cell viability and proliferation (CellTiter 96^®^ Aqueous; Promega Charbonnières les Bains, France) following 5 days of exposure to IH, SH or N. Cells were seeded at 5000 cells/well in 96-well plates 24 h before exposure. HepG2 cells were incubated with a medium (control), 1 µM acriflavine, 5 µg/mL pazopanib or 5 µM macitentan and exposed to IH, SH or N on day 1. On day 5, cells were incubated with MTT for two hours, followed by OD measurement at 570 nm using a spectrophotometer, according to the provider’s instructions.

### 4.7. Transwell Migration Assays 

The migration properties of HepG2 cells were assessed in a transwell migration experiment. Briefly, 24-well transwell chambers with 8 mm pore polycarbonate filters (Sigma Aldrich, St. Louis, MI, USA) were coated with type I collagen (Sigma Aldrich, St. Louis, MI, USA). In total, 100,000 HepG2 cells diluted in DMEM without any supplements were introduced into the upper part of each transwell chamber, and 600 µL of the DMEM medium supplemented with 10% FBS was placed in the well constituting the lower part of the chamber as a cell chemoattractant. After 2 days of normoxia, intermittent hypoxia, or sustained hypoxia exposure, the medium was removed and cells that had not migrated, i.e., still therefore present on the upper layer of the transwell, were gently removed using dedicated cotton swabs. Cells attached to the lower layer of the transwell were fixed with absolute ethanol, stained using 0.2% crystal violet for 10 min at room temperature, and then imaged using a camera-coupled optical microscope (×10 magnification Olympus CK2; Olympus, USA). Migration ability was estimated by OD measurement (595 nm) after solubilization in acetic acid. The results were obtained from 3 independent experiments and compared with cells exposed to normoxia. 

### 4.8. Transcript Expression Evaluation 

Dry HepG2 cell pellets from normoxia, intermittent hypoxia and sustained hypoxia were used to perform RT-q-PCR. Total RNA extraction was performed with NucleoSpin RNA, following the manufacturer recommendations, with RNase-free DNase treatment (Macherey Nagel, Düren, Germany). cDNA was reverse transcribed from 1 µg of total RNA with the SuperScriptIII First-Strand Synthesis followed by RNaseH (1 µL) treatment (LifeTechnologies, Carlsbad, CA, USA). Real-time PCR was conducted using the QuantiTect SYBR Green RT-PCR kit (Qiagen, Venlo, The Netherlands) and the Stratagene 3005MxPro (Santa Clara, CA, USA). The primers (LifeTechnologies, Carlsbad, CA, USA) were all used at 400 nM, Tm at 60 °C (for more detail, see the Appendix A). 

### 4.9. ET-1 and VEGF Level Assessments by ELISA 

ET-1 cytosolic protein produced by HepG2 cells was evaluated using an endothelin-1 ELISA kit on total cellular extracts (Enzo Life Sciences, New York, NY, USA). VEGF levels secreted by HepG2 cells were evaluated using a human VEGF Duoset ELISA kit (R&D system, Minneapolis, MN, USA) on HepG2 culture supernatants according to the supplier’s recommendations.

### 4.10. Statistical Analysis

Continuous data are expressed as the mean ± SD. Data were analyzed by repeated-measures analysis of variance (ANOVA), followed by paired *t*-test for 2 × 2 comparisons. As inequality of variance could not be excluded, a Greenhouse–Geisser adjustment was used. We tested the effects of time, oxygen exposure and treatments, as well as the interactions between all these factors. Homogeneous data were analyzed using fixed-effect models, and heterogeneous data were analyzed using random effect size models. We considered data as heterogeneous when their I² values were superior to 50%. 

Two-sided significance tests were used. We considered *p*-values < 0.05 as significant, corrected by Bonferroni’s method for multiple comparison. All statistical analyses were performed using SPSS statistics 20.0 for Windows (IBM corp., Chicago, IL, USA). 

## Figures and Tables

**Figure 1 ijms-24-06875-f001:**
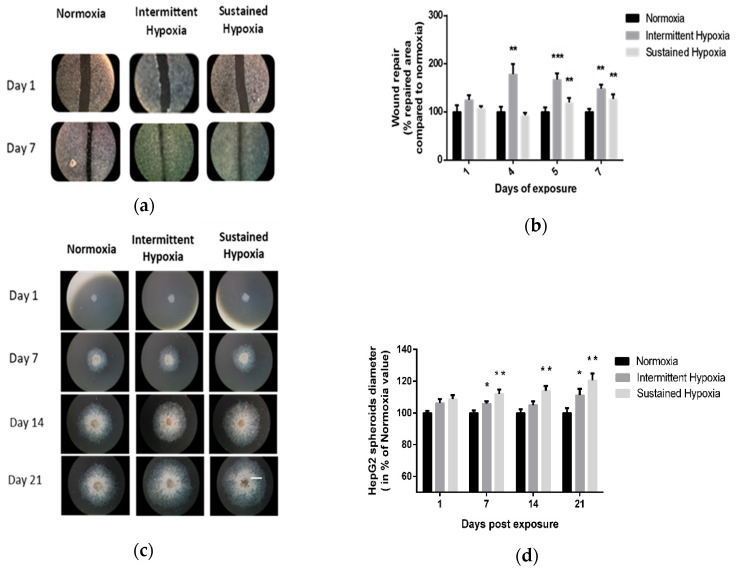
In vitro sustained hypoxia and intermittent hypoxia increase hepatic tumor cell expansion. (**a**) Representative illustrations of HepG2 cell invasiveness in 2D, assessed by wound healing, before and after 7 days of normoxia, intermittent hypoxia or sustained hypoxia exposure (white scale bar = 600 µm). (**b**) Wound healing, expressed as a % of repaired area compared with normoxia, of HepG2 cells exposed to 7 days of normoxia, intermittent hypoxia or sustained hypoxia; *n* = 5 experiments per group with at least 3 wells/experiment. Intermittent hypoxia global effect *p* < 0.00001, ** *p* < 0.01 on D4 and D7 for IH vs. N and on D5 and D7 for SH vs. N and *** *p* < 0.001 for IH vs. N on D5, one-way repeated measures ANOVA. Sustained hypoxia global effect *p* = 0.006, one-way repeated measures ANOVA and ** *p* < 0.01 on D5 and on day 7. (**c**) Representative illustrations of HepG2 spheroid expansion before and after 5, 7, 14 and 21 days of normoxia, intermittent hypoxia or sustained hypoxia exposure (white scale bar = 1000 µm). (**d**) HepG2 spheroid expansion in response to 21 days of intermittent hypoxia; *n* = 5 experiments per group (6 to 12 wells/experiment). * *p* < 0.05, repeated measures ANOVA). Post hoc analysis showed significant differences * *p* < 0.05 on days 7 and 21 of exposure. HepG2 spheroid expansion in response to 21 days of sustained hypoxia; *n* = 5 experiments per group (6 to 12 wells/experiment). * *p* < 0.01, repeated measures ANOVA. Post hoc analysis showed significant differences * *p* < 0.05 on days 7, 14 and 21 of exposure.

**Figure 2 ijms-24-06875-f002:**
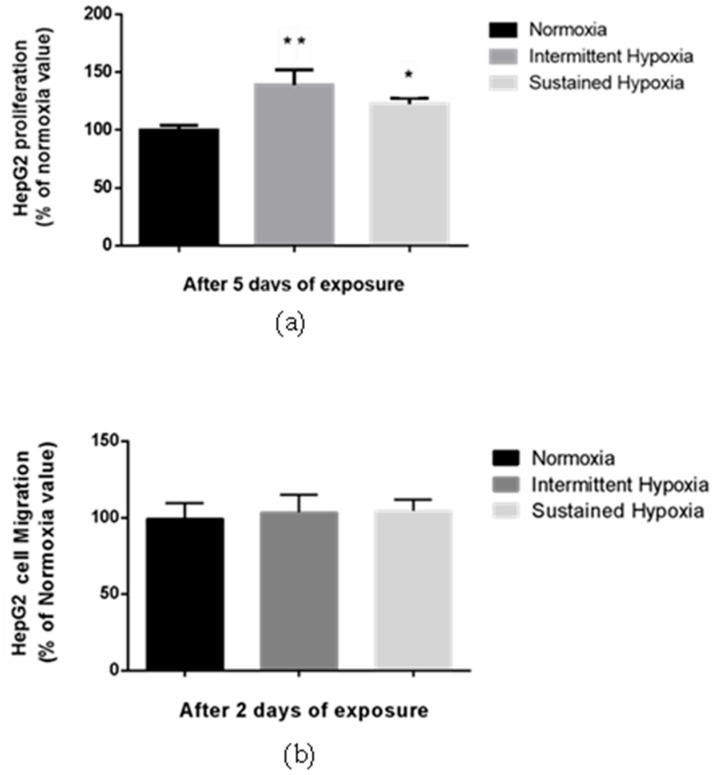
Sustained hypoxia and intermittent hypoxia increase cell proliferation in vitro. (**a**) Proliferation, expressed as a % of normoxia values, of viable HepG2 cells quantified by MTT staining after 5 days of normoxia, intermittent hypoxia or sustained hypoxia exposure; *n* = 3 independent experiments/group (at least 18 wells/group). * *p* < 0.05 and ** *p* < 0.01 on D5 for IH cells compared with normoxia cells, and *p* < 0.05 for SH cells compared with N cells, Mann–Whitney U test. (**b**) Migration, expressed as a % of normoxia values, of HepG2 cells after 2 days of normoxia, intermittent hypoxia or sustained hypoxia exposure; *n* = 3 independent experiment/group (6 wells/group: 2 transwells/experiments). Mann–Whitney U test.

**Figure 3 ijms-24-06875-f003:**
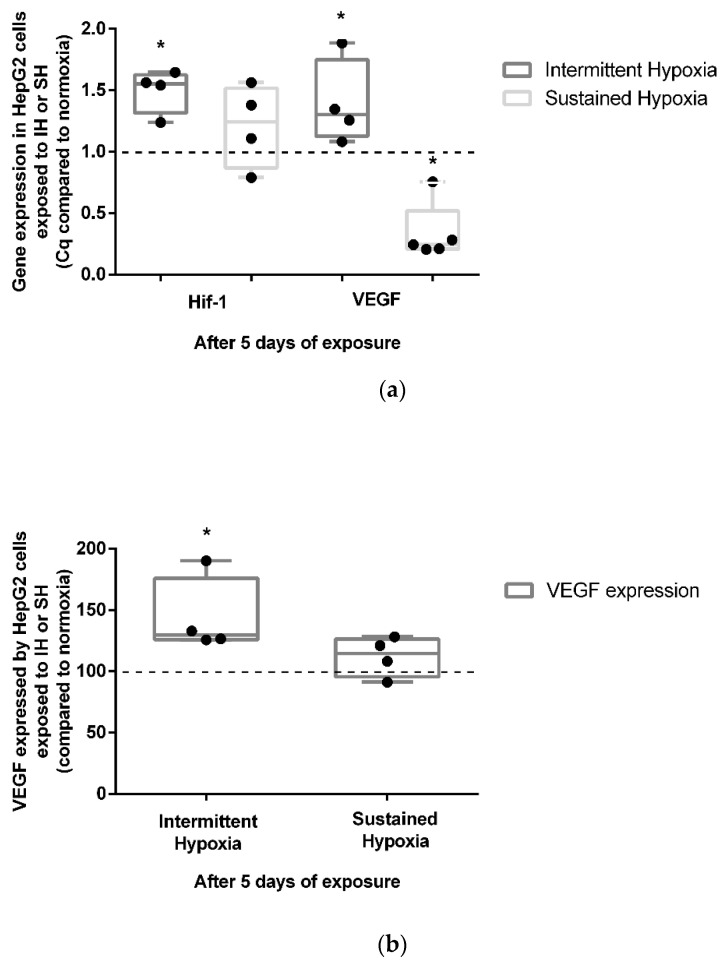
In vitro effects of IH and SH on HIF-1α gene expression, and VEGF gene and protein expression. (**a**) Intermittent hypoxia is associated with an increase in HIF-1α and VEGF gene expressions in HepG2 cells after 5 days of exposure. The levels of HIF-1α and VEGF gene expression were measured by RT-QPCR; *n* = 3 independent experiments/group. * *p* < 0.05, Mann–Whitney U test. (**b**) VEGF expression increases in HepG2 cells after 5 days of IH but not SH exposure. The levels of VEGF protein in the cell’s supernatant were measured by ELISA; *n* = 3 independent experiments/group. * *p* < 0.05) Mann–Whitney U test.

**Figure 4 ijms-24-06875-f004:**
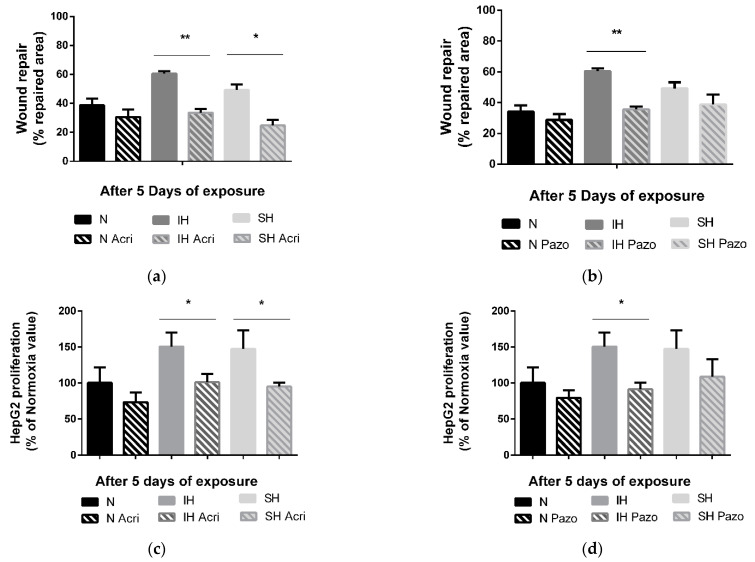
Effects of acriflavine and pazopanib on wound healing and proliferation under sustained hypoxia or intermittent hypoxia. (**a**) Wound healing, expressed as % of repaired area, of HepG2 cells exposed to 5 days of normoxia (N), intermittent hypoxia (IH) or sustained hypoxia (SH) and treated or not by acriflavine (Acri); *n* = 3 experiments per group with at least 3 wells/experiment. ** *p* < 0.01 on D5 for IH acriflavine-treated cells compared with IH untreated and * *p* < 0.05 on D5 for SH acriflavine-treated cells compared with SH untreated cells, Mann–Whitney U test. (**b**) Wound healing, expressed as a % of repaired area, of HepG2 cells exposed to 5 days of normoxia, intermittent hypoxia or sustained hypoxia and treated or not by pazopanib (Pazo); *n* = 3 experiments per group with at least 3 wells/experiment. ** *p* < 0.01 on D5 for IH pazopanib-treated cells compared with IH untreated, Mann–Whitney U test. (**c**) Proliferation, expressed as a % of Normoxia values, of viable HepG2 cells quantified by MTT staining after 5 days of normoxia, intermittent hypoxia or sustained hypoxia and treated or not by acriflavine (Acri); *n* = 3 independent experiments/group, 6 to 12 wells/group: at least 2 wells/experiment. * *p* < 0.05 on D5 for IH acriflavine-treated cells compared with IH untreated, *p* < 0.05 for CH acriflavine-treated cells compared with CH untreated cells, Mann–Whitney U test. (**d**) Proliferation, expressed as a % of control values (respectively N, IH or CH), of viable HepG2 cells quantified by MTT staining after 5 days of normoxia, intermittent hypoxia or sustained hypoxia and treated or not by pazopanib (Pazo); *n* = at least 3 independent experiments/group (9 wells/group: at least 2 wells/experiment). * *p* < 0.05 on D5 for IH pazopanib-treated cells compared with IH untreated, Mann–Whitney U test.

## Data Availability

Not applicable.

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
