# Peer review of "Differential Impact of Intermittent vs. Sustained Hypoxia on HIF-1, VEGF and Proliferation of HepG2 Cells"

_ijms, 2023, doi:10.3390/ijms24086875_

Round 1

Reviewer 1 Report

Clear introduction and rationale

Clear methods and valid statistical analysis

Results support the role of IH (major manifestation of OSA) in amplifying the proliferation of hepatic cancer cells through VEGF and HIF-1α in-vitro

Main comment:

In few lines, discuss the differences in the pathological mechanisms between IH and SH, particularly in cancer progression.

Some references to help:

1.    Saxena K, Jolly MK. Acute vs. Chronic vs. Cyclic Hypoxia: Their Differential Dynamics, Molecular Mechanisms, and Effects on Tumor Progression. Biomolecules. 2019 Aug 3;9(8):339. doi: 10.3390/biom9080339. PMID: 31382593; PMCID: PMC6722594.

2.    Hui AS, Striet JB, Gudelsky G, Soukhova GK, Gozal E, Beitner-Johnson D, Guo SZ, Sachleben LR Jr, Haycock JW, Gozal D, Czyzyk-Krzeska MF. Regulation of catecholamines by sustained and intermittent hypoxia in neuroendocrine cells and sympathetic neurons. Hypertension. 2003 Dec;42(6):1130-6. doi: 10.1161/01.HYP.0000101691.12358.26. Epub 2003 Nov 3. PMID: 14597643.

other minor comments:

·        Needs comprehensive academic English proofreading

·        Please unify abbreviations (ex: OSA vs OSAS)

Good luck

Author Response

We thank the referee for her/his positive appreciation of our manuscript.

Main comment:

In few lines, discuss the differences in the pathological mechanisms between IH and SH, particularly in cancer progression.

We thank the referee for suggesting additional references. We have added a paragraph in the discussion section to further discuss the differences between IH and SH:

Line 251: “This differential expression of HIF-1 could thus explain why IH, but not SH, was able to increase VEGF expression in our study.” and 256 to 262 p8: “Among the pathophysiological mechanisms involved in cancer occurrence and progression, intermittent hypoxia has been shown to enhance angiogenesis, inflammation, stem cell renewal and tumor cell invasion and metastasis compared to sustained hypoxia[1]. Interestingly, most of these effects are mediated by HIF-1 stabilization. Severity (O2 level) and frequency of hypoxia-reoxygenation cycles, the main parameters controlling HIF-1 stabilization, are thus crucial determinants of the cellular impact of hypoxia and could explain differential impact between SH and IH.[1].”

 other minor comments:

  • Needs comprehensive academic English proofreading

Thank you for your comment. We have followed your recommendation so English proofreading was performed all along the manuscript.

  • Please unify abbreviations (ex: OSA vs OSAS)

All abreviations were carefully checked and OSA was used everywhere.

Bibliography of the response to the referee

[1]        K. Saxena et al., ‘Acute vs. Chronic vs. Cyclic Hypoxia: Their Differential Dynamics, Molecular Mechanisms, and Effects on Tumor Progression.’, Biomolecules, vol. 9, no. 8, p. 339, Aug. 2019, doi: 10.3390/biom9080339.

Reviewer 2 Report

  Reviewer comments and suggestions

The authors aim was to compare the respective consequences of intermittent and sustained hypoxia on HIF-1, endothelin-1 and VEGF expression, cell proliferation and migration in HepG2 liver tumor cells. The study measured HIF-1α, endothelin-1 and VEGF expression at the mRNA and/or protein levels. Study tested inhibitors of HIF-1 (acriflavine), endothelin-1 (macitentan) and VEGF (pazopanib) on HepG2 cells. 

HIF-1 and VEGF, but not endothelin-1, expressions were increased in IH but not SH. Both IH and SH stimulate hepatic cancer cells proliferation via distinct signaling pathways. The authors finally suggested that the relationship could be understandable in OSA patients with cancer, enhancing tumor progression.

Overall, the manuscript was well written. However, a few concerns/comments needed to be explained/modified. 

  1. Line 36 Need to discuss the meaning apneas and hypopneas (the author can mention the feature of diagnosing or characterization) 
  2. Line 46 Needs to discuss the confounders” owing to many confounders altering 46 the robustness of the findings”
  3. Line 86 Why only emphasize this only? s “on HIF-1, VEGF, 86 and ET-1”.
  4. Line 134 Is it okay to represent like this (percentage)
  5. Line 168 Do you mention why VEGF shows these differences
  6. Line 214 how the author connects wound healing with cancer proliferation
  7. Line 270-271 Please make it simple for clear understanding for your common reader.
  8. Line 274 The author did not discuss the pathways.
  9. Line 301 better to add basic points related to these “HIF-1, VEGF and ET-1”
  10.  Please check References 36,51 , 54 and 55 are these references present in the MS

Author Response

 We thank the referee for her/his positive appreciation of our manuscript. The comments are addressed below. Our comments are avaible below in blue script (and in red the version track change of the manuscript).

  1. Line 36 Need to discuss the meaning apneas and hypopneas (the author can mention the feature of diagnosing or characterization) 

The following sentence was added in the introduction, lines 36-39: “Apneas are defined as a decrease in airflow measured by nasal pressure of more than 90% from baseline lasting at least 10 seconds. Hypopnoeas are usually defined as a decrease in airflow of more than 30% with a decrease in blood oxygen saturation of more than 3% and an associated microarousal)”.

2. Line 48 Needs to discuss the confounders” owing to many confounders altering the robustness of the findings”

The paragraph was modified as follows :

Line 49, p2: “However, many confounders alter the robustness of these findings, such as patient characteristics (gender, obesity, age, smoking or menopausal status), variability in cancer sites, cancer cell type and susceptibility to IH, and unknown temporality between OSA and cancer incidence. Consequently, the link between OSA and cancer needs to be addressed by studies investigating specific types of cancer, as recently pointed out by Gozal et al [2].”

3. Line 88 Why only emphasize this only? s “on HIF-1, VEGF, 86 and ET-1”.

As mentioned in the introduction section lines 70 to 75, p2 : HIF-1 is a key factor regulating cancer progression under hypoxic conditions [1], [3] and is very well described for its role in mediating physiopathological consequences of OSA [4], [5]  as well as some target genes including VEGF[6]–[8] and ET-1 [4], [9]  Among its target genes, VEGF and ET-1 are also activated in IH and are well-known to promote tumor cell proliferation and metastasis [8], [10]–[12]. Thus, they were good candidates for our mechanistic investigation of the differential impact of SH and IH on tumor cells.

4. Line 133-134 Is it okay to represent like this (percentage)

We decided to express the IH results as a percentage of normoxic controls, as is often applied in in vitro studies, in order to facilitate reading and to account for variability between independent experiments.

5. Line 168 Do you mention why VEGF shows these differences

In our study, VEGF seems implicated in IH but not SH. We hypothesized that differential expression and stabilization of HIF-1 under these two different hypoxia regimens [1] leads to different activation of VEGF target gene.

We added a sentence in the discussion section, line 251, p 8: “This differential expression of HIF-1 could thus explain why IH, but not SH, was able to increase VEGF expression in our study.”

6.Line 218 how the author connects wound healing with cancer proliferation.

Wound healing relies on two cellular processes: cell proliferation and cell migration. Since we did not observe any impact of hypoxia on cell migration, we conclude that the differences in wound healing are explained by the increased cell proliferation which is one of the main mechanisms involved in cancer progression. So, we clarified this concept and added a paragraph in the discussion line 216 to 223: “We observed that both IH and SH induced an increase in wound healing and spheroid expansion, associated with progression and invasiveness of tumor cells. Wound healing reflects two cellular processes: cell proliferation and cell migration. Our results showed an increase in tumour cell proliferation, but not in cell migration, under SH and IH conditions. This suggests that the differences in wound healing could be explained by increased cell proliferation which is one of the main mechanisms involved in cancer progression.”

7. Line 281 Please make it simple for clear understanding for your common reader.

We modified the paragraph p8 line 282 as follows: “In OSA patients with tumors, systemic IH due to apneas could add to the effects of local SH in the tumor micro-environment. We can hypothesize that both hypoxic conditions participate in the increased cancer progression observed in OSA patients and in animal models of OSA.”

8. Line 290 The author did not discuss the pathways.

Line 278 p 8 : We modified the sentence accordingly : “To conclude, our study shows that while both intermittent and sustained hypoxia have the ability to enhance liver tumor development in vitro, the tumor-promoting effect of intermittent hypoxia are more specifically mediated by HIF-1 and its target gene VEGF.”

9. Line 317 better to add basic points related to these “HIF-1, VEGF and ET-1”

We modified this paragraph accordingly, line 318 p 9:

« The potential mechanisms behind the effects of IH and SH were investigated using three pharmacological inhibitors:

-acriflavine, a HIF-1 inhibitor [13],

-pazopanib (Sigma Aldrich, USA) an inhibitor of VEGF receptor tyrosine kinase activity [14], macitentan, a non-selective ET-1 receptor antagonist [15] (graciously provided by Actelion Pharmaceuticals, Switzerland). »

10. Please check References 36,51 , 54 and 55 are these references present in the MS

Reference 36 was added in the manuscript. Reference 51 is cited in the materials&methods section, line 303. References 54 and 55 were removed from the bibliography list. We apologize for these mistakes.

Bibliography of the response to referees

[1]        K. Saxena et al., ‘Acute vs. Chronic vs. Cyclic Hypoxia: Their Differential Dynamics, Molecular Mechanisms, and Effects on Tumor Progression.’, Biomolecules, vol. 9, no. 8, p. 339, Aug. 2019, doi: 10.3390/biom9080339.

[2]        D. Gozal, I. Almendros, A. I. Phipps, F. Campos-Rodriguez, M. A. Martínez-García, and R. Farré, ‘Sleep apnoea adverse effects on cancer: True, false, or too many confounders?’, Int J Mol Sci, vol. 21, no. 22, pp. 1–21, Nov. 2020, doi: 10.3390/ijms21228779.

[3]        G. L. Semenza, ‘Hypoxia-inducible factors: Mediators of cancer progression and targets for cancer therapy’, Trends Pharmacol Sci, vol. 33, no. 4, pp. 207–214, 2012, doi: 10.1016/j.tips.2012.01.005.

[4]        E. Gras et al., ‘Endothelin-1 mediates intermittent hypoxia-induced inflammatory vascular remodeling through HIF-1 activation’, J Appl Physiol, vol. 120, no. 4, pp. 437–443, Feb. 2016, doi: 10.1152/japplphysiol.00641.2015.

[5]        E. Belaidi, J. Morand, E. Gras, J.-L. Pépin, and D. Godin-Ribuot, ‘Targeting the ROS-HIF-1-endothelin axis as a therapeutic approach for the treatment of obstructive sleep apnea-related cardiovascular complications.’, Pharmacol Ther, vol. 168, pp. 1–11, Dec. 2016, doi: 10.1016/j.pharmthera.2016.07.010.

[6]        A. Briançon-Marjollet, M. Henri, J.-L. Pépin, E. Lemarié, P. Lévy, and R. Tamisier, ‘Altered in vitro Endothelial Repair and Monocyte Migration in Obstructive Sleep Apnea: Implication of VEGF and CRP’, Sleep, vol. 37, no. 11, pp. 1825–1832, Nov. 2014, doi: 10.5665/sleep.4180.

[7]        R. Schulz, C. Hummel, S. Heinemann, W. Seeger, and F. Grimminger, ‘Serum levels of vascular endothelial growth factor are elevated in patients with obstructive sleep apnea and severe nighttime hypoxia.’, Am J Respir Crit Care Med, vol. 165, no. 1, pp. 67–70, Jan. 2002, doi: 10.1164/ajrccm.165.1.2101062.

[8]        D. W. Yoon et al., ‘Accelerated tumor growth under intermittent hypoxia is associated with hypoxia-inducible factor-1-dependent adaptive responses to hypoxia’, Oncotarget, vol. 8, no. 37, pp. 61592–61603, Sep. 2017, doi: 10.18632/oncotarget.18644.

[9]        A. Briançon-Marjollet et al., ‘Endothelin regulates intermittent hypoxia-induced lipolytic remodelling of adipose tissue and phosphorylation of hormone-sensitive lipase.’, J Physiol, vol. 594, no. 6, pp. 1727–40, Mar. 2016, doi: 10.1113/JP271321.

[10]      M. Minoves et al., ‘Chronic intermittent hypoxia, a hallmark of obstructive sleep apnea, promotes 4T1 breast cancer development through endothelin-1 receptors.’, Sci Rep, vol. 12, no. 1, p. 12916, Jul. 2022, doi: 10.1038/s41598-022-15541-8.

[11]      I. Almendros et al., ‘Obesity and intermittent hypoxia increase tumor growth in a mouse model of sleep apnea’, Sleep Med, vol. 13, no. 10, pp. 1254–1260, 2012, doi: 10.1016/j.sleep.2012.08.012.

[12]      I. Almendros and D. Gozal, ‘Intermittent hypoxia and cancer: Undesirable bed partners?’, Respir Physiol Neurobiol, vol. 256, pp. 79–86, Oct. 2018, doi: 10.1016/j.resp.2017.08.008.

[13]      K. Lee, H. Zhang, D. Z. Qian, S. Rey, J. O. Liu, and G. L. Semenza, ‘Acriflavine inhibits HIF-1 dimerization, tumor growth, and vascularization’, Proceedings of the National Academy of Sciences, vol. 106, no. 42, pp. 17910–17915, Oct. 2009, doi: 10.1073/pnas.0909353106.

[14]      Y. Jia, J. Zhang, J. Feng, F. Xu, H. Pan, and W. Xu, ‘Design, synthesis and biological evaluation of pazopanib derivatives as antitumor agents.’, Chem Biol Drug Des, vol. 83, no. 3, pp. 306–16, Mar. 2014, doi: 10.1111/cbdd.12243.

[15]      M. Clozel, ‘Endothelin research and the discovery of macitentan for the treatment of pulmonary arterial hypertension.’, Am J Physiol Regul Integr Comp Physiol, vol. 311, no. 4, pp. R721–R726, 2016, doi: 10.1152/ajpregu.00475.2015.
